# FastSpeech 2: Fast and High-Quality End-to-End Text to Speech

**Yi Ren**[1]*, **Chenxu Hu**[1]*, **Xu Tan**[2], **Tao Qin**[2], **Sheng Zhao**[3], **Zhou Zhao**[1]†, **Tie-Yan Liu**[2]

[1]Zhejiang University
{rayeren,chenxuhu,zhaozhou}@zju.edu.cn

[2]Microsoft Research Asia
{xuta,taoqin,tyliu}@microsoft.com

[3]Microsoft Azure Speech
Sheng.Zhao@microsoft.com

## ABSTRACT

Non-autoregressive text to speech (TTS) models such as FastSpeech (Ren et al., 2019) can synthesize speech significantly faster than previous autoregressive models with comparable quality. The training of FastSpeech model relies on an autoregressive teacher model for duration prediction (to provide more information as input) and knowledge distillation (to simplify the data distribution in output), which can ease the one-to-many mapping problem (i.e., multiple speech variations correspond to the same text) in TTS. However, FastSpeech has several disadvantages: 1) the teacher-student distillation pipeline is complicated and time-consuming, 2) the duration extracted from the teacher model is not accurate enough, and the target mel-spectrograms distilled from teacher model suffer from information loss due to data simplification, both of which limit the voice quality. In this paper, we propose FastSpeech 2, which addresses the issues in FastSpeech and better solves the one-to-many mapping problem in TTS by 1) directly training the model with ground-truth target instead of the simplified output from teacher, and 2) introducing more variation information of speech (e.g., pitch, energy and more accurate duration) as conditional inputs. Specifically, we extract duration, pitch and energy from speech waveform and directly take them as conditional inputs in training and use predicted values in inference. We further design FastSpeech 2s, which is the first attempt to directly generate speech waveform from text in parallel, enjoying the benefit of fully end-to-end inference. Experimental results show that 1) FastSpeech 2 achieves a 3x training speed-up over FastSpeech, and FastSpeech 2s enjoys even faster inference speed; 2) FastSpeech 2 and 2s outperform FastSpeech in voice quality, and FastSpeech 2 can even surpass autoregressive models. Audio samples are available at https://speechresearch.github.io/fastspeech2/.

## 1 INTRODUCTION

Neural network based text to speech (TTS) has made rapid progress and attracted a lot of attention in the machine learning and speech community in recent years (Wang et al., 2017; Shen et al., 2018; Ming et al., 2016; Arik et al., 2017; Ping et al., 2018; Ren et al., 2019; Li et al., 2019). Previous neural TTS models (Wang et al., 2017; Shen et al., 2018; Ping et al., 2018; Li et al., 2019) first generate mel-spectrograms *autoregressively* from text and then synthesize speech from the generated mel-spectrograms using a separately trained vocoder (Van Den Oord et al., 2016; Oord et al., 2017; Prenger et al., 2019; Kim et al., 2018; Yamamoto et al., 2020; Kumar et al.,

---

*Authors contribute equally to this work.
†Corresponding author

2019). They usually suffer from slow inference speed and robustness (word skipping and repeating) issues (Ren et al., 2019; Chen et al., 2020). In recent years, non-autoregressive TTS models (Ren et al., 2019; Łańcucki, 2020; Kim et al., 2020; Lim et al., 2020; Miao et al., 2020; Peng et al., 2019) are designed to address these issues, which generate mel-spectrograms with extremely fast speed and avoid robustness issues, while achieving comparable voice quality with previous autoregressive models.

Among those non-autoregressive TTS methods, FastSpeech (Ren et al., 2019) is one of the most successful models. FastSpeech designs two ways to alleviate the one-to-many mapping problem: 1) Reducing data variance in the target side by using the generated mel-spectrogram from an autoregressive teacher model as the training target (i.e., knowledge distillation). 2) Introducing the duration information (extracted from the attention map of the teacher model) to expand the text sequence to match the length of the mel-spectrogram sequence. While these designs in FastSpeech ease the learning of the one-to-many mapping problem (see Section 2.1) in TTS, they also bring several disadvantages: 1) The two-stage teacher-student training pipeline makes the training process complicated. 2) The target mel-spectrograms generated from the teacher model have some information loss[1] compared with the ground-truth ones, since the quality of the audio synthesized from the generated mel-spectrograms is usually worse than that from the ground-truth ones. 3) The duration extracted from the attention map of teacher model is not accurate enough.

In this work, we propose FastSpeech 2 to address the issues in FastSpeech and better handle the one-to-many mapping problem in non-autoregressive TTS. To simplify the training pipeline and avoid the information loss due to data simplification in teacher-student distillation, we directly train the FastSpeech 2 model with ground-truth target instead of the simplified output from a teacher. To reduce the information gap (input does not contain all the information to predict the target) between the input (text sequence) and target output (mel-spectrograms) and alleviate the one-to-many mapping problem for non-autoregressive TTS model training, we introduce some variation information of speech including pitch, energy and more accurate duration into FastSpeech: in training, we extract duration, pitch and energy from the target speech waveform and directly take them as conditional inputs; in inference, we use values predicted by the predictors that are jointly trained with the FastSpeech 2 model. Considering the pitch is important for the prosody of speech and is also difficult to predict due to the large fluctuations along time, we convert the pitch contour into pitch spectrogram using continuous wavelet transform (Tuteur, 1988; Grossmann & Morlet, 1984) and predict the pitch in the frequency domain, which can improve the accuracy of predicted pitch. To further simplify the speech synthesis pipeline, we introduce FastSpeech 2s, which does not use mel-spectrograms as intermediate output and directly generates speech waveform from text in inference, enjoying low latency in inference. Experiments on the LJSpeech (Ito, 2017) dataset show that 1) FastSpeech 2 enjoys much simpler training pipeline (3x training time reduction) than FastSpeech while inherits its advantages of fast, robust and controllable (even more controllable in pitch and energy) speech synthesis, and FastSpeech 2s enjoys even faster inference speed; 2) FastSpeech 2 and 2s outperform FastSpeech in voice quality, and FastSpeech 2 can even surpass autoregressive models. We attach audio samples generated by FastSpeech 2 and 2s at `https://speechresearch.github.io/fastspeech2/`.

The main contributions of this work are summarized as follows:

- FastSpeech 2 achieves a 3x training speed-up over FastSpeech by simplifying the training pipeline.
- FastSpeech 2 alleviates the one-to-many mapping problem in TTS and achieves better voice quality.
- FastSpeech 2s further simplifies the inference pipeline for speech synthesis while maintaining high voice quality, by directly generating speech waveform from text.

## 2 FASTSPEECH 2 AND 2S

In this section, we first describe the motivation of the design in FastSpeech 2, and then introduce the architecture of FastSpeech 2, which aims to improve FastSpeech to better handle the one-to-

---

[1]The speech generated by the teacher model loses some variation information about pitch, energy, prosody, etc., and is much simpler and less diverse than the original recording in the training data.

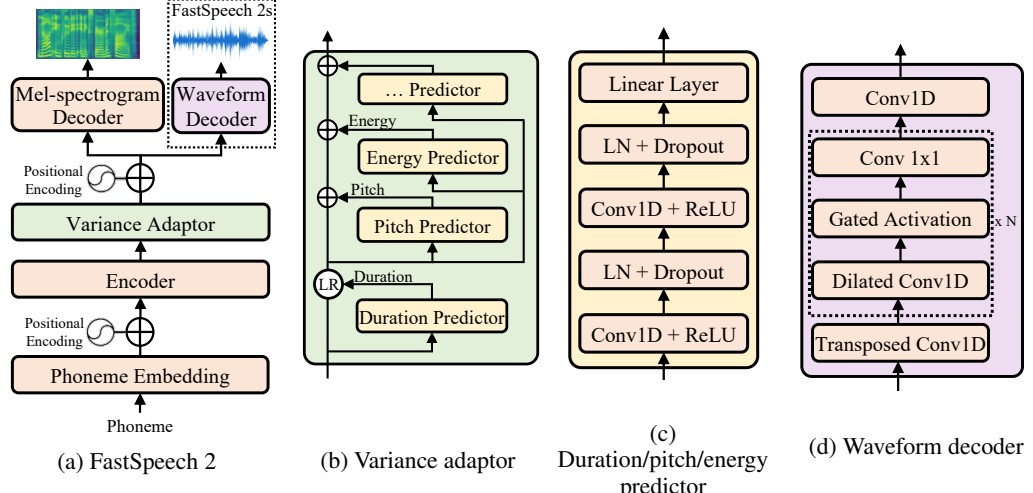

| (a) FastSpeech 2 | (b) Variance adaptor | (c) Duration/pitch/energy predictor | (d) Waveform decoder |

Figure 1: The overall architecture for FastSpeech 2 and 2s. LR in subfigure (b) denotes the length regulator proposed in FastSpeech. LN in subfigure (c) denotes layer normalization.

many mapping problem, with simpler training pipeline and higher voice quality. At last, we extend FastSpeech 2 to FastSpeech 2s for fully end-to-end text-to-waveform synthesis[2].

## 2.1 MOTIVATION

TTS is a typical one-to-many mapping problem (Wang et al., 2017; Zhu et al., 2017; Jayne et al., 2012; Gadermayr et al., 2020; Chen et al., 2021), since multiple possible speech sequences can correspond to a text sequence due to variations in speech, such as pitch, duration, sound volume and prosody. In non-autoregressive TTS, the only input information is text which is not enough to fully predict the variance in speech. In this case, the model is prone to overfit to the variations of the target speech in the training set, resulting in poor generalization ability. As mentioned in Section 1, although FastSpeech designs two ways to alleviate the one-to-many mapping problem, they also bring about several issues including 1) the complicated training pipeline; 2) information loss of target mel-spectrogram as analyzed in Table 1; and 3) not accurate enough ground-truth duration as shown in Table 5a. In the following subsection, we introduce the detailed design of FastSpeech 2 which aims to address these issues.

## 2.2 MODEL OVERVIEW

The overall model architecture of FastSpeech 2 is shown in Figure 1a. The encoder converts the phoneme embedding sequence into the phoneme hidden sequence, and then the variance adaptor adds different variance information such as duration, pitch and energy into the hidden sequence, finally the mel-spectrogram decoder converts the adapted hidden sequence into mel-spectrogram sequence in parallel. We use the feed-forward Transformer block, which is a stack of self-attention (Vaswani et al., 2017) layer and 1D-convolution as in FastSpeech (Ren et al., 2019), as the basic structure for the encoder and mel-spectrogram decoder. Different from FastSpeech that relies on a teacher-student distillation pipeline and the phoneme duration from a teacher model, Fast-Speech 2 makes several improvements. First, we remove the teacher-student distillation pipeline, and directly use ground-truth mel-spectrograms as target for model training, which can avoid the information loss in distilled mel-spectrograms and increase the upper bound of the voice quality. Second, our variance adaptor consists of not only duration predictor but also pitch and energy predictors, where 1) the duration predictor uses the phoneme duration obtained by forced alignment (McAuliffe et al., 2017) as training target, which is more accurate than that extracted from the attention map of autoregressive teacher model as verified experimentally in Section 3.2.2; and 2) the

---

[2]In this work, text-to-waveform refers to phoneme-to-waveform, while our method can also be appied to character-level sequence directly.

additional pitch and energy predictors can provide more variance information, which is important to ease the one-to-many mapping problem in TTS. Third, to further simplify the training pipeline and push it towards a fully end-to-end system, we propose FastSpeech 2s, which directly generates waveform from text, without cascaded mel-spectrogram generation (acoustic model) and waveform generation (vocoder). In the following subsections, we describe detailed designs of the variance adaptor and direct waveform generation in our method.

## 2.3 VARIANCE ADAPTOR

The variance adaptor aims to add variance information (e.g., duration, pitch, energy, etc.) to the phoneme hidden sequence, which can provide enough information to predict variant speech for the one-to-many mapping problem in TTS. We briefly introduce the variance information as follows: 1) phoneme duration, which represents how long the speech voice sounds; 2) pitch, which is a key feature to convey emotions and greatly affects the speech prosody; 3) energy, which indicates frame-level magnitude of mel-spectrograms and directly affects the volume and prosody of speech. More variance information can be added in the variance adaptor, such as emotion, style and speaker, and we leave it for future work. Correspondingly, the variance adaptor consists of 1) a duration predictor (i.e., the length regulator, as used in FastSpeech), 2) a pitch predictor, and 3) an energy predictor, as shown in Figure 1b. In training, we take the ground-truth value of duration, pitch and energy extracted from the recordings as input into the hidden sequence to predict the target speech. At the same time, we use the ground-truth duration, pitch and energy as targets to train the duration, pitch and energy predictors, which are used in inference to synthesize target speech. As shown in Figure 1c, the duration, pitch and energy predictors share similar model structure (but different model parameters), which consists of a 2-layer 1D-convolutional network with ReLU activation, each followed by the layer normalization and the dropout layer, and an extra linear layer to project the hidden states into the output sequence. In the following paragraphs, we describe the details of the three predictors respectively.

**Duration Predictor**    The duration predictor takes the phoneme hidden sequence as input and predicts the duration of each phoneme, which represents how many mel frames correspond to this phoneme, and is converted into logarithmic domain for ease of prediction. The duration predictor is optimized with mean square error (MSE) loss, taking the extracted duration as training target. Instead of extracting the phoneme duration using a pre-trained autoregressive TTS model in FastSpeech, we use Montreal forced alignment (MFA) (McAuliffe et al., 2017) tool[3] to extract the phoneme duration, in order to improve the alignment accuracy and thus reduce the information gap between the model input and output.

**Pitch Predictor**    Previous neural network based TTS systems with pitch prediction (Arik et al., 2017; Gibiansky et al., 2017) often predict pitch contour directly. However, due to high variations of ground-truth pitch, the distribution of predicted pitch values is very different from ground-truth distribution, as analyzed in Section 3.2.2. To better predict the variations in pitch contour, we use continuous wavelet transform (CWT) to decompose the continuous pitch series into pitch spectrogram (Suni et al., 2013; Hirose & Tao, 2015) and take the pitch spectrogram as the training target for the pitch predictor which is optimized with MSE loss. In inference, the pitch predictor predicts the pitch spectrogram, which is further converted back into pitch contour using inverse continuous wavelet transform (iCWT). We describe the details of pitch extraction, CWT, iCWT and pitch predictor architecture in Appendix D. To take the pitch contour as input in both training and inference, we quantize pitch $F_0$ (ground-truth/predicted value for train/inference respectively) of each frame to 256 possible values in log-scale and further convert it into pitch embedding vector $p$ and add it to the expanded hidden sequence.

**Energy Predictor**    We compute L2-norm of the amplitude of each short-time Fourier transform (STFT) frame as the energy. Then we quantize energy of each frame to 256 possible values uniformly, encoded it into energy embedding $e$ and add it to the expanded hidden sequence similarly to

---

[3]MFA is an open-source system for speech-text alignment with good performance, which can be trained on paired text-audio corpus without any manual alignment annotations. We train MFA on our training set only without other external dataset. We will work on non-autoregressive TTS without external alignment models in the future.

pitch. We use an energy predictor to predict the original values of energy instead of the quantized values and optimize the energy predictor with MSE loss[4].

## 2.4 FASTSPEECH 2S

To enable fully end-to-end text-to-waveform generation, in this subsection, we extend FastSpeech 2 to FastSpeech 2s, which directly generates waveform from text, without cascaded mel-spectrogram generation (acoustic model) and waveform generation (vocoder). As shown in Figure 1a, FastSpeech 2s generates waveform conditioning on intermediate hidden, which makes it more compact in inference by discarding mel-spectrogram decoder and achieve comparable performance with a cascaded system. We first discuss the challenges in non-autoregressive text-to-waveform generation, then describe details of FastSpeech 2s, including model structure and training and inference processes.

**Challenges in Text-to-Waveform Generation**   When pushing TTS pipeline towards fully end-to-end framework, there are several challenges: 1) Since the waveform contains more variance information (e.g., phase) than mel-spectrograms, the information gap between the input and output is larger than that in text-to-spectrogram generation. 2) It is difficult to train on the audio clip that corresponds to the full text sequence due to the extremely long waveform samples and limited GPU memory. As a result, we can only train on a short audio clip that corresponds to a partial text sequence which makes it hard for the model to capture the relationship among phonemes in different partial text sequences and thus harms the text feature extraction.

**Our Method**   To tackle the challenges above, we make several designs in the waveform decoder: 1) Considering that the phase information is difficult to predict using a variance predictor (Engel et al., 2020), we introduce adversarial training in the waveform decoder to force it to implicitly recover the phase information by itself (Yamamoto et al., 2020). 2) We leverage the mel-spectrogram decoder of FastSpeech 2, which is trained on the full text sequence to help on the text feature extraction. As shown in Figure 1d, the waveform decoder is based on the structure of WaveNet (Van Den Oord et al., 2016) including non-causal convolutions and gated activation (Van den Oord et al., 2016). The waveform decoder takes a sliced hidden sequence corresponding to a short audio clip as input and upsamples it with transposed 1D-convolution to match the length of audio clip. The discriminator in the adversarial training adopts the same structure in Parallel WaveGAN (Yamamoto et al., 2020) which consists of ten layers of non-causal dilated 1-D convolutions with leaky ReLU activation function. The waveform decoder is optimized by the multi-resolution STFT loss and the LSGAN discriminator loss following Parallel WaveGAN. In inference, we discard the mel-spectrogram decoder and only use the waveform decoder to synthesize speech audio.

## 2.5 DISCUSSIONS

In this subsection, we discuss how FastSpeech 2 and 2s differentiate from previous and concurrent works.

Compared with Deep Voice (Arik et al., 2017), Deep Voice 2 (Gibiansky et al., 2017) and other methods Fan et al. (2014); Ze et al. (2013) which generate waveform autoregressively and also predict variance information such as duration and pitch, Fastspeech 2 and 2s adopt self-attention based feed-forward network to generate mel-spectrograms or waveform in parallel. While some existing non-autoregressive acoustic models (Zeng et al., 2020; Lim et al., 2020; Kim et al., 2020) mostly focus on improving the duration accuracy, FastSpeech 2 and 2s provide more variation information (duration, pitch and energy) as inputs to reduce the information gap between the input and output. A concurrent work (Łańcucki, 2020) employs pitch prediction in phoneme level, while FastSpeech 2 and 2s predict more fine-grained pitch contour in frame level. In addition, to improve the prosody in synthesized speech, FastSpeech 2 and 2s further introduce continuous wavelet transform to model the variations in pitch.

While some text-to-waveform models such as ClariNet (Ping et al., 2019) jointly train an autoregressive acoustic model and a non-autoregressive vocoder, FastSpeech 2s embraces the fully non-autoregressive architecture for fast inference. A concurrent work called EATS (Donahue et al., 2020)

---

[4]We do not transform energy using CWT since energy is not as highly variable as pitch on LJSpeech dataset, and we do not observe gains when using it.

also employs non-autoregressive architecture and adversarial training to convert text to waveform directly and mainly focuses on predicting the duration of each phoneme end-to-end using a differentiable monotonic interpolation scheme. Compared with EATS, FastSpeech 2s additionally provides more variation information to ease the one-to-many mapping problem in TTS.

Previous non-autoregressive vocoders (Oord et al., 2017; Prenger et al., 2019; Yamamoto et al., 2020; Kumar et al., 2019) are not complete text-to-speech systems, since they convert time aligned linguistic features to waveforms, and require a separate linguistic model to convert input text to linguistic features or an acoustic model to convert input text to acoustic features (e.g., mel-spectrograms). FastSpeech 2s is the first attempt to directly generate waveform from phoneme sequence fully in parallel, instead of linguistic features or mel-spectrograms.

## 3 EXPERIMENTS AND RESULTS

### 3.1 EXPERIMENTAL SETUP

**Datasets** We evaluate FastSpeech 2 and 2s on LJSpeech dataset (Ito, 2017). LJSpeech contains 13,100 English audio clips (about 24 hours) and corresponding text transcripts. We split the dataset into three sets: 12,228 samples for training, 349 samples (with document title LJ003) for validation and 523 samples (with document title LJ001 and LJ002) for testing. For subjective evaluation, we randomly choose 100 samples in test set. To alleviate the mispronunciation problem, we convert the text sequence into the phoneme sequence (Arik et al., 2017; Wang et al., 2017; Shen et al., 2018; Sun et al., 2019) with an open-source grapheme-to-phoneme tool[5]. We transform the raw waveform into mel-spectrograms following Shen et al. (2018) and set frame size and hop size to 1024 and 256 with respect to the sample rate 22050.

**Model Configuration** Our FastSpeech 2 consists of 4 feed-forward Transformer (FFT) blocks (Ren et al., 2019) in the encoder and the mel-spectrogram decoder. The output linear layer in the decoder converts the hidden states into 80-dimensional mel-spectrograms and our model is optimized with mean absolute error (MAE). We add more detailed configurations of FastSpeech 2 and 2s used in our experiments in Appendix A. The details of training and inference are added in Appendix B.

### 3.2 RESULTS

| Method | MOS |
|---|---|
| *GT* | $4.30 \pm 0.07$ |
| *GT (Mel + PWG)* | $3.92 \pm 0.08$ |
| *Tacotron 2 (Shen et al., 2018) (Mel + PWG)* | $3.70 \pm 0.08$ |
| *Transformer TTS (Li et al., 2019) (Mel + PWG)* | $3.72 \pm 0.07$ |
| *FastSpeech (Ren et al., 2019) (Mel + PWG)* | $3.68 \pm 0.09$ |
| *FastSpeech 2 (Mel + PWG)* | $3.83 \pm 0.08$ |
| *FastSpeech 2s* | $3.71 \pm 0.09$ |

(a) The MOS with 95% confidence intervals.

| Method | CMOS |
|---|---|
| *FastSpeech 2* | 0.000 |
| *FastSpeech* | -0.885 |
| *Transformer TTS* | -0.235 |

(b) CMOS comparison.

Table 1: Audio quality comparison.

In this section, we first evaluate the audio quality, training and inference speedup of FastSpeech 2 and 2s. Then we conduct analyses and ablation studies of our method[6].

#### 3.2.1 MODEL PERFORMANCE

**Audio Quality** To evaluate the perceptual quality, we perform mean opinion score (MOS) (Chu & Peng, 2006) evaluation on the test set. Twenty native English speakers are asked to make quality

---

[5]https://github.com/Kyubyong/g2p
[6]We put some audio samples in the supplementary materials and https://speechresearch.github.io/fastspeech2/.

| Method | Training Time (h) | Inference Speed (RTF) | Inference Speedup |
|---|---|---|---|
| *Transformer TTS (Li et al., 2019)* | 38.64 | $9.32 \times 10^{-1}$ | / |
| *FastSpeech (Ren et al., 2019)* | 53.12 | $1.92 \times 10^{-2}$ | $48.5\times$ |
| *FastSpeech 2* | **17.02** | $1.95 \times 10^{-2}$ | $47.8\times$ |
| *FastSpeech 2s* | 92.18 | $\mathbf{1.80 \times 10^{-2}}$ | $\mathbf{51.8\times}$ |

Table 2: The comparison of training time and inference latency in waveform synthesis. The training time of *FastSpeech* includes teacher and student training. RTF denotes the real-time factor, that is the time (in seconds) required for the system to synthesize one second waveform. The training and inference latency tests are conducted on a server with 36 Intel Xeon CPUs, 256GB memory, 1 NVIDIA V100 GPU and batch size of 48 for training and 1 for inference. Besides, we do not include the time of GPU memory garbage collection and transferring input and output data between the CPU and the GPU. The speedup in waveform synthesis for FastSpeech is larger than that reported in Ren et al. (2019) since we use Parallel WaveGAN as the vocoder which is much faster than WaveGlow.

judgments about the synthesized speech samples. The text content keeps consistent among different systems so that all testers only examine the audio quality without other interference factors. We compare the MOS of the audio samples generated by *FastSpeech 2* and *FastSpeech 2s* with other systems, including 1) *GT*, the ground-truth recordings; 2) *GT (Mel + PWG)*, where we first convert the ground-truth audio into mel-spectrograms, and then convert the mel-spectrograms back to audio using Parallel WaveGAN (Yamamoto et al., 2020) (PWG); 3) *Tacotron 2 (Shen et al., 2018) (Mel + PWG)*; 4) *Transformer TTS (Li et al., 2019) (Mel + PWG)*; 5) *FastSpeech (Ren et al., 2019) (Mel + PWG)*. All the systems in 3), 4) and 5) use Parallel WaveGAN as the vocoder for a fair comparison. The results are shown in Table 1. It can be seen that FastSpeech 2 can surpass and FastSpeech 2s can match the voice quality of autoregressive models *Transformer TTS* and *Tacotron 2*. Importantly, FastSpeech 2 outperforms FastSpeech, which demonstrates the effectiveness of providing variance information such as pitch, energy and more accurate duration and directly taking ground-truth speech as training target without using teacher-student distillation pipeline.

**Training and Inference Speedup**  FastSpeech 2 simplifies the training pipeline of FastSpeech by removing the teacher-student distillation process, and thus reduces the training time. We list the total training time of *Transformer TTS* (the autoregressive teacher model), FastSpeech (including the training of *Transformer TTS* teacher model and *FastSpeech* student model) and *FastSpeech 2* in Table 2. It can be seen that FastSpeech 2 reduces the total training time by $3.12\times$ compared with FastSpeech. Note that training time here only includes acoustic model training, without considering the vocoder training. Therefore, we do not compare the training time of FastSpeech 2s here. We then evaluate the inference latency of FastSpeech 2 and 2s compared with the autoregressive Transformer TTS model, which has the similar number of model parameters with FastSpeech 2 and 2s. We show the inference speedup for waveform generation in Table 2. It can be seen that compared with the Transformer TTS model, FastSpeech 2 and 2s speeds up the audio generation by $47.8\times$ and $51.8\times$ respectively in waveform synthesis. We can also see that FastSpeech 2s is faster than FastSpeech 2 due to fully end-to-end generation.

### 3.2.2   ANALYSES ON VARIANCE INFORMATION

| Method | $\sigma$ | $\gamma$ | $\mathcal{K}$ | $DTW$ |
|---|---|---|---|---|
| *GT* | 54.4 | 0.836 | 0.977 | / |
| *Tacotron 2* | 44.1 | 1.28 | 1.311 | 26.32 |
| *TransformerTTS* | 40.8 | 0.703 | 1.419 | 24.40 |
| *FastSpeech* | 50.8 | 0.724 | -0.041 | 24.89 |
| *FastSpeech 2* | **54.1** | 0.881 | **0.996** | 24.39 |
| *FastSpeech 2 - CWT* | 42.3 | 0.771 | 1.115 | 25.13 |
| *FastSpeech 2s* | 53.9 | **0.872** | 0.998 | **24.37** |

Table 3: Standard deviation ($\sigma$), skewness ($\gamma$), kurtosis ($\mathcal{K}$) and average DTW distances (DTW) of pitch in ground-truth and synthesized audio.

**More Accurate Variance Information in Synthesized Speech**    In the paragraph, we measure if providing more variance information (e.g., pitch and energy) as input in FastSpeech 2 and 2s can indeed synthesize speech with more accurate pitch and energy.

For pitch, we compute the moments (standard deviation ($\sigma$), skewness ($\gamma$) and kurtosis ($\mathcal{K}$)) (Andreeva et al., 2014; Niebuhr & Skarnitzl, 2019) and average dynamic time warping (DTW) Müller (2007) distance of the pitch distribution for the ground-truth speech and synthesized speech. The results are shown in Table 3. It can be seen that compared with FastSpeech, the moments ($\sigma$, $\gamma$ and $\mathcal{K}$) of generated audio of FastSpeech 2/2s are more close to the ground-truth audio and the average DTW distances to the ground-truth pitch are smaller than other methods, demonstrating that FastSpeech 2/2s can generate speech with more natural pitch contour (which can result in better prosody) than FastSpeech. We also conduct a case study on generated pitch contours in Appendix D.

| Method | FastSpeech | FastSpeech 2 | FastSpeech 2s |
|--------|-----------|--------------|---------------|
| MAE | 0.142 | 0.131 | 0.133 |

Table 4: The mean absolute error (MAE) of the energy in synthesized speech audio.

For energy, we compute the mean absolute error (MAE) between the frame-wise energy extracted from the generated waveform and the ground-truth speech. To ensure that the numbers of frames in the synthesized and ground-truth speech are the same, we use the ground-truth duration extracted by MFA in both FastSpeech and FastSpeech 2. The results are shown in Table 4. We can see that the MAE of the energy for FastSpeech 2/2s are smaller than that for FastSpeech, indicating that they both synthesize speech audio with more similar energy to the ground-truth audio.

**More Accurate Duration for Model Training**    We then analyze the accuracy of the provided duration information to train the duration predictor and the effectiveness of more accurate duration for better voice quality based on FastSpeech. We manually align 50 audio generated by the teacher model and the corresponding text in phoneme level and get the ground-truth phoneme-level duration. We compute the average of absolute phoneme boundary differences (McAuliffe et al., 2017) using the duration from the teacher model of FastSpeech and from MFA as used in this paper respectively. The results are shown in Table 5a. We can see that MFA can generate more accurate duration than the teacher model of FastSpeech. Next, we replace the duration used in FastSpeech (from teacher model) with that extracted by MFA, and conduct the CMOS (Loizou, 2011) test to compare the voice quality between the two FastSpeech models trained with different durations[7]. The results are listed in Table 5b and it can be seen that more accurate duration information improves the voice quality of FastSpeech, which verifies the effectiveness of our improved duration from MFA.

| Method | $\Delta$ (ms) |
|--------|---------------|
| Duration from teacher model | 19.68 |
| Duration from MFA | 12.47 |

| Setting | CMOS |
|---------|------|
| FastSpeech + Duration from teacher | 0 |
| FastSpeech + Duration from MFA | +0.195 |

(a) Alignment accuracy comparison.          (b) CMOS comparison.

Table 5: The comparison of the duration from teacher model and MFA. $\Delta$ means the average of absolute boundary differences.

### 3.2.3    ABLATION STUDY

**Pitch and Energy Input**    We conduct ablation studies to demonstrate the effectiveness of several variance information of FastSpeech 2 and 2s, including pitch and energy[8]. We conduct CMOS evaluation for these ablation studies. The results are shown in Table 6. We find that removing the energy (Row 3 in both subtables) in FastSpeech 2 and 2s results in performance drop in terms of voice quality (-0.040 and -0.160 CMOS respectively), indicating that energy is effective for FastSpeech 2 in improving the voice quality, and more effective for FastSpeech 2s. We also find that removing the

---

[7]Both models are trained with mel-spectrograms generated by the teacher model.

[8]We do not study duration information since duration is a necessary for FastSpeech and FastSpeech 2. Besides, we have already analyzed the effectiveness of our improved duration in the last paragraph.

pitch (Row 4 in both subtables) in FastSpeech 2 and 2s results in -0.245 and -1.130 CMOS respectively, which demonstrates the effectiveness of pitch. When we remove both pitch and energy (the last row in both subtables), the voice quality further drops, indicating that both pitch and energy can help improve the performance of FastSpeech 2 and 2s.

**Predicting Pitch in Frequency Domain**   To study the effectiveness of predicting pitch in frequency domain using continuous wavelet transform (CWT) as described in Section 2.3, we directly fit the pitch contour with mean square error like energy in FastSpeech 2 and 2s. We conduct CMOS evaluation and get CMOS drops of 0.185 and 0.201 for FastSpeech 2 and 2s respectively. We also compute the moments of pitch and average DTW distance to the ground-truth pitch as shown in row 6 (denoeted as *FastSpeech 2 - CWT*) in Table 3. The results demonstrate that CWT can help model the pitch better and improve the prosody of synthesized speech, and thus obtaining better CMOS score.

**Mel-Spectrogram Decoder in FastSpeech 2s**   To verify the effectiveness of the mel-spectrogram decoder in FastSpeech 2s on text feature extraction as described in Section 2.4, we remove the mel-spectrogram decoder and conduct CMOS evaluation. It causes a 0.285 CMOS drop, which demonstrates that the mel-spectrogram decoder is essential to high-quality waveform generation.

| Setting | CMOS |
|---|---|
| *FastSpeech 2* | 0 |
| *FastSpeech 2 - energy* | -0.040 |
| *FastSpeech 2 - pitch* | -0.245 |
| *FastSpeech 2 - pitch - energy* | -0.370 |

| Setting | CMOS |
|---|---|
| *FastSpeech 2s* | 0 |
| *FastSpeech 2s - energy* | -0.160 |
| *FastSpeech 2s - pitch* | -1.130 |
| *FastSpeech 2s - pitch - energy* | -1.355 |

(a) CMOS comparison for FastSpeech 2.     (b) CMOS comparison for FastSpeech 2s.

Table 6: CMOS comparison in the ablation studies.

## 4   CONCLUSION

In this work, we proposed FastSpeech 2, a fast and high-quality end-to-end TTS system, to address the issues in FastSpeech and ease the one-to-many mapping problem: 1) we directly train the model with ground-truth mel-spectrograms to simplify the training pipeline and also avoid information loss compared with FastSpeech; and 2) we improve the duration accuracy and introduce more variance information including pitch and energy to ease the one-to-many mapping problem, and improve pitch prediction by introducing continuous wavelet transform. Moreover, based on FastSpeech 2, we further developed FastSpeech 2s, a non-autoregressive text-to-waveform generation model, which enjoys the benefit of fully end-to-end inference and achieves faster inference speed. Our experimental results show that FastSpeech 2 and 2s outperform FastSpeech, and FastSpeech 2 can even surpass autoregressive models in terms of voice quality, with much simpler training pipeline while inheriting the advantages of fast, robust and controllable speech synthesis of FastSpeech.

High quality, fast and fully end-to-end training without any external libraries is definitely the ultimate goal of neural TTS and also a very challenging problem. To ensure high quality of FastSpeech 2, we use an external high-performance alignment tool and pitch extraction tools, which may seem a little complicated, but are very helpful for high-quality and fast speech synthesis. We believe there will be more simpler solutions to achieve this goal in the future and we will certainly work on fully end-to-end TTS without external alignment models and tools. We will also consider more variance information (Zhang et al., 2021) to further improve the voice quality and speed up the inference with more light-weight model (Luo et al., 2021).

ACKNOWLEDGMENTS

This work was supported in part by the National Key R&D Program of China (Grant No.2018AAA0100603), National Natural Science Foundation of China (Grant No.62072397), Zhejiang Natural Science Foundation (Grant No.LR19F020006), National Natural Science Foundation of China (Grant No.61836002) and X Lab, the Second Academy of CASIC, Beijing, 100854, China.

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

## A  MODEL CONFIGURATION

Our FastSpeech 2 consists of 4 feed-forward Transformer (FFT) blocks (Ren et al., 2019) in the encoder and the mel-spectrogram decoder. In each FFT block, the dimension of phoneme embeddings and the hidden size of the self-attention are set to 256. The number of attention heads is set to 2 and the kernel sizes of the 1D-convolution in the 2-layer convolutional network after the self-attention layer are set to 9 and 1, with input/output size of 256/1024 for the first layer and 1024/256 in the second layer. The size of the phoneme vocabulary is 76, including punctuations. In the variance predictor, the kernel sizes of the 1D-convolution are set to 3, with input/output sizes of 256/256 for both layers and the dropout rate is set to 0.5. Our waveform decoder consists of 1-layer transposed 1D-convolution with filter size 64 and 30 dilated residual convolution blocks, whose skip channel size and kernel size of 1D-convolution are set to 64 and 3. The configurations of the discriminator in FastSpeech 2s are the same as Parallel WaveGAN (Yamamoto et al., 2020). We list hyperparameters and configurations of all models used in our experiments in Table 7.

| Hyperparameter | Transformer TTS | FastSpeech/FastSpeech 2/2s |
|---|---|---|
| Phoneme Embedding Dimension | 256 | 256 |
| Pre-net Layers | 3 | / |
| Pre-net Hidden | 256 | / |
| Encoder Layers | 4 | 4 |
| Encoder Hidden | 256 | 256 |
| Encoder Conv1D Kernel | 9 | 9 |
| Encoder Conv1D Filter Size | 1024 | 1024 |
| Encoder Attention Heads | 2 | 2 |
| Mel-Spectrogram Decoder Layers | 4 | 4 |
| Mel-Spectrogram Decoder Hidden | 256 | 256 |
| Mel-Spectrogram Decoder Conv1D Kernel | 9 | 9 |
| Mel-Spectrogram Decoder Conv1D Filter Size | 1024 | 1024 |
| Mel-Spectrogram Decoder Attention Headers | 2 | 2 |
| Encoder/Decoder Dropout | 0.1 | 0.1 |
| Variance Predictor Conv1D Kernel | / | 3 |
| Variance Predictor Conv1D Filter Size | / | 256 |
| Variance Predictor Dropout | / | 0.5 |
| Waveform Decoder Convolution Blocks | / | 30 |
| Waveform Decoder Dilated Conv1D Kernel size | / | 3 |
| Waveform Decoder Transposed Conv1D Filter Size | / | 64 |
| Waveform Decoder Skip Channlel Size | / | 64 |
| Batch Size | 48 | 48/48/12 |
| Total Number of Parameters | 24M | 23M/27M/28M |

Table 7: Hyperparameters of Transformer TTS, FastSpeech and FastSpeech 2/2s.

## B  TRAINING AND INFERENCE

We train FastSpeech 2 on 1 NVIDIA V100 GPU, with batchsize of 48 sentences. We use the Adam optimizer (Kingma & Ba, 2014) with $\beta_1 = 0.9$, $\beta_2 = 0.98$, $\varepsilon = 10^{-9}$ and follow the same learning rate schedule in Vaswani et al. (2017). It takes 160k steps for training until convergence. In the inference process, the output mel-spectrograms of our FastSpeech 2 are transformed into audio samples using pre-trained Parallel WaveGAN (Yamamoto et al., 2020)[9]. For FastSpeech 2s, we train the model on 2 NVIDIA V100 GPUs, with batchsize of 6 sentences on each GPU. The waveform decoder takes the sliced hidden states corresponding to 20,480 waveform sample clips as input. The optimizer and learning rate schedule for FastSpeech 2s are the same as FastSpeech 2. The details of the adversarial training follow Parallel WaveGAN (Yamamoto et al., 2020). It takes 600k steps for training until convergence for FastSpeech 2s.

---

[9]https://github.com/kan-bayashi/ParallelWaveGAN

## C  MODELING PITCH WITH CONTINUOUS WAVELET TRANSFORM

### C.1  CONTINUOUS WAVELET TRANSFORM

Given a contfnous pitch contour function $F_0$, we can convert it to pitch spectrogram $W(\tau, t)$ using continuous wavelet transform (Tuteur, 1988; Grossmann & Morlet, 1984):

$$W(\tau, t) = \tau^{-1/2} \int_{-\infty}^{+\infty} F_0(x)\psi(\frac{x-t}{\tau})dx$$

where $\psi$ is the Mexican hat mother wavelet (Ryan, 1994), $F_0(x)$ is the pitch value in position $x$, $\tau$ and $t$ are scale and position of wavelet respectively. The original pitch contour $F_0$ can be recovered from the wavelet representation $W(\tau, t)$ by inverse continuous wavelet transform (iCWT) using the following formula:

$$F_0(t) = \int_{-\infty}^{+\infty} \int_{0}^{+\infty} W(\tau, t)\, \tau^{-5/2} \psi\left(\frac{x-t}{\tau}\right) dx d\tau$$

Suppose that we decompose the pitch contour $F_0$ into 10 scales (Ming et al., 2016), $F_0$ can be represented by 10 separate components given by:

$$W_i(t) = W(2^{i+1}\tau_0, t)(i + 2.5)^{-5/2} \tag{1}$$

where $i = 1, ..., 10$ and $\tau_0 = 5ms$, which is originally proposed in Suni et al. (2013). Given 10 wavelet components $\hat{W}_i(t)$, we can recompose pitch contour $\hat{F}_0$ by the following formula (Ming et al., 2016):

$$\hat{F}_0(t) = \sum_{i=1}^{10} \hat{W}_i(t)(i + 2.5)^{-5/2} \tag{2}$$

### C.2  IMPLEMENTATION DETAILS

First we extract the pitch contour using PyWorldVocoder[10]. Since CWT is very sensitive to discontinuous signals, we preprocess the pitch contour as follows: 1) we use linear interpolation to fill the unvoiced frame in pitch contour; 2) we transform the resulting pitch contour to logarithmic scale; 3) we normalize it to zero mean and unit variance for each utterance, and we have to save the original utterance-level mean and variance for pitch contour reconstruction; and 4) we convert the normalized pitch contour to pitch spectrogram using continuous wavelet transform following Equation 1.

As shown in Figure 2, pitch predictor consists of a 2-layer 1D-convolutional network with ReLU activation, each followed by the layer normalization and the dropout layer, and an extra linear layer to project the hidden states into the pitch spectrogram. To predict the mean/variance of recovered pitch contour for each utterance, we average the hidden states output by the 1D-convolutional network on the time dimension to a global vector and project it to mean and variance using a linear layer.

We train the pitch predictor with ground-truth pitch spectrogram and the mean/variance of pitch contour and optimize it with mean square error. During inference, we predict the pitch spectrogram and the mean/variance of recovered pitch contour using pitch predictor, inverse the pitch spectrogram to pitch contour with inverse continuous wavelet transform (iCWT) following Equation 2, and finally denormalize it with the predicted mean/variance.

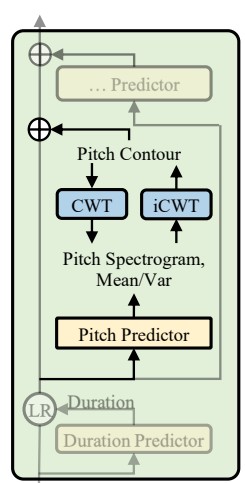

Figure 2: Details in pitch predictor. CWT and iCWT denote continuous wavelet transform and inverse continuous wavelet transform respectively.

---

[10]https://github.com/JeremyCCHsu/Python-Wrapper-for-World-Vocoder

# D    CASE STUDY ON PITCH CONTOUR

In this section, we conduct the case study on pitch contours of the audios generated by different methods. We randomly choose 1 utterance from the test set and plot the pitch countor of ground-truth audio samples and that generated by *FastSpeech*, *FastSpeech 2*, *FastSpeech 2s* in Figure 3. We can see that FastSpeech 2 and 2s can capture the variations in pitch better than FastSpeech thanks to taking pitch information as input.

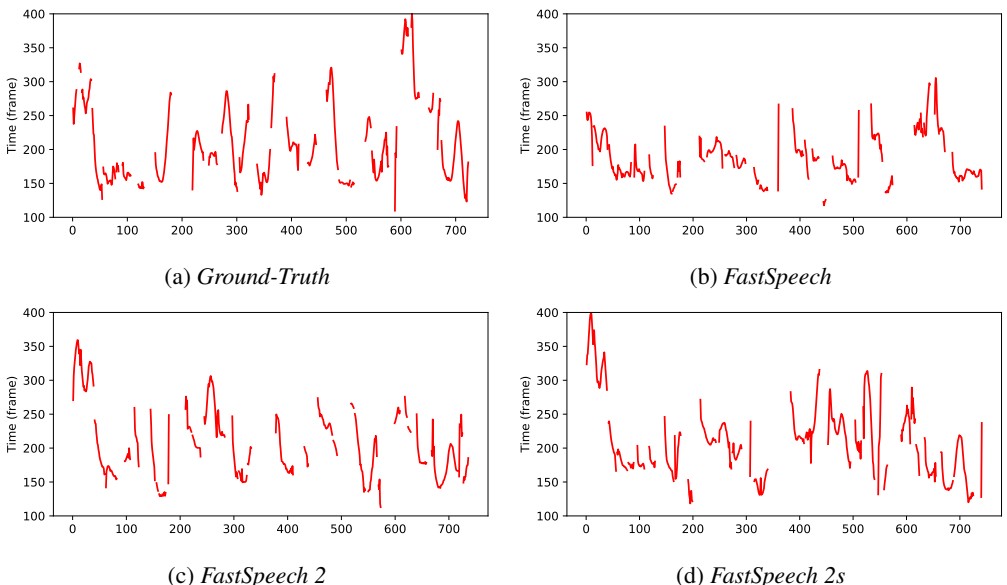

(a) *Ground-Truth*

(b) *FastSpeech*

(c) *FastSpeech 2*

(d) *FastSpeech 2s*

Figure 3: Pitch contours extracted from generated and ground-truth audio samples. We only plot the voiced part of pitch contour. The input text is "*The worst, which perhaps was the English, was a terrible falling-off from the work of the earlier presses*".

# E    VARIANCE CONTROL

FastSpeech 2 and 2s introduce several variance information to ease the one-to-many mapping problem in TTS. As a byproduct, they also make the synthesized speech more controllable and can be used to manually control pitch, duration and energy (volume) of synthesized audio. As a demonstration, we manipulate pitch input to control the pitch of synthesized speech in this subsubsection. We show the mel-spectrograms before and after the pitch manipulation in Figure 4. From the samples, we can see that FastSpeech 2 generates high-quality mel-spectrograms after adjusting the $\hat{F}_0$ from 0.75 to 1.50 times. Such manipulation can also be applied to FastSpeech 2s and the results are put in the supplementary materials. We also put the audio samples controlled by other variance information in supplementary materials.

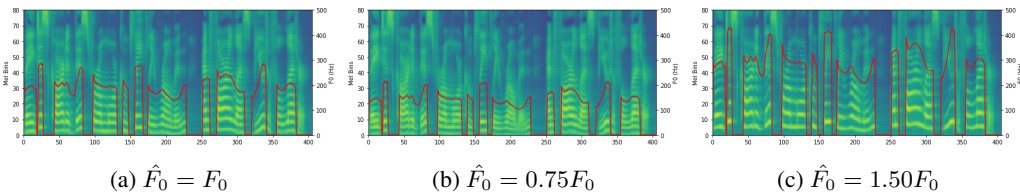

(a) $\hat{F}_0 = F_0$

(b) $\hat{F}_0 = 0.75 F_0$

(c) $\hat{F}_0 = 1.50 F_0$

Figure 4: The mel-spectrograms of the voice with different $\hat{F}_0$. $F_0$ is the fundamental frequency of original audio. The red curves denote $\hat{F}_0$ contours. The input text is "*They discarded this for a more completely Roman and far less beautiful letter.*"

