# OpenReview forum: "FastSpeech 2: Fast and High-Quality End-to-End Text to Speech"
_ICLR.cc/2021/Conference — ICLR 2021 Poster_

### Official Review · AnonReviewer3 · 2020-10-27
**Fastspeech 2: fast TTS training and inference with decent speech quality**

**Rating:** 7
**Confidence:** 4

**Review:**

Summary:
In FastSpeech 2 and 2s, the authors propose several changes to the non-autoregressive FastSpeech model. An autoregressive teacher model was used in FastSpeech to alleviate the one-to-many problem of TTS by providing phoneme durations estimated by attention mechanism, and predicted mel-spectrograms as soft targets to the final non-autoregressive model. In FastSpeech 2, the authors do away with the teacher-student knowledge distillation pipeline and instead utilize the following to alleviate the one-to-many problem while retaining a non-autoregressive architecture:
- ground-truth mel spectrogram is used as the training targets
- a force alignment tool is used to extract more accurate phoneme durations
- pitch and energy information is also used as conditioning for the model as a means to introduce variance information

These changes boost the speech quality scores of the FastSpeech 2 architecture over autoregressive TTS architectures while reducing both training and inference times considerably. The authors also introduce an end-to-end text-to-waveform variant of the model: FastSpeech 2s. This model leverages the mel-spectrogram decoder, adversarial training, and a wavenet-like architecture to enable waveform prediction at a shorter time-scale. The Fastspeech 2s architecture performs at par with autoregressive TTS models. The authors further perform ablation studies that demonstrate the importance of utilizing accurate duration, pitch and energy information.

Positives:
1. The authors explain the model architecture clearly and perform thorough evaluation and ablation studies.
2. The changes in FastSpeech lead to improvements in performance both in terms of speech quality and time (training and inference). The quality of the samples provided is pretty good and since the models having only been trained with a publicly available dataset which is not unreasonably large shows great potential.

Negatives:
1. The authors make several claims that need substantiating. Below are a few:
    - 2.3: The authors claim that pitch is the key feature to convey emotion and affect speech prosody while energy affects speech volume. This is not entirely correct. Emotion can be conveyed by variation in pitch as well as energy and the contribution of each of these features towards speech prosody cannot be so easily disentangled. I would recommend adding that energy is also an important factor in speech prosody.
    - footnote 3: The authors mention that energy is not as highly variable as speech. This may be an artifact of the dataset being utilized which consists only of neutral speech. This could be included in the text as well.
    - 3.2.2: The authors claim that since the distribution of pitch extracted from FastSpeech 2 synthesized speech is closer to groung truth distribution, the pitch contours are more natural. This claim is easier made if a comparison of time-series pitch contours is made. Including a simple DTW distance would strengthen the argument. Additionally, there are metrics proposed in prosody transfer literature[1] that can be used here.
2. FastSpeech 2 method section is not very clearly explained. While it is understandable that there are space constraints, each component requires at least one sentence explaining its purpose and how it works. The paragraph lacks any detail of the adversarial training method and how phase retrieval happens.

Minor edits:
"challenges lie in" -> challenges in

References:
[1] Skerry-Ryan, R. J., et al. "Towards end-to-end prosody transfer for expressive speech synthesis with tacotron." arXiv preprint arXiv:1803.09047 (2018).

---

> ### Author Response · Authors · 2020-11-22
> **Response to AnonReviewer3**
>
> Thanks for your comments on our paper. We reply to your questions as follows:
>
> **[About the importance of energy]**
> Thanks for your advice! We highlight the importance of energy in modeling the prosody in the new version of the paper.
>
> **[About the variability of energy in other datasets]**
> Yes, you are right.  We update the footnote [4] to “We do not transform energy using CWT since energy is not as highly variable as pitch **on LJSpeech dataset**, and we do not observe gains when using it.”
>
> **[About the DTW distances of pitch]**
> Thanks for your advice. We add the DTW distances for each model to Table 3. The results show that FastSpeech 2 and 2s can generate more accurate pitch than other models, which is consistent with our analysis.
>
> **[About more details in FastSpeech 2s]**
> We guess you mean FastSpeech 2s method section. For adversarial training, the waveform decoder and discriminator in FastSpeech 2s adopt the same structure of generator and discriminator in Parallel WaveGAN respectively and are optimized with the losses in LSGAN *[1]*. After the adversarial training, the waveform decoder can generate high-fidelity waveform, which is the phase retrieval process. We add these details of the adversarial training and phase retrieval methods in the new version of the paper.
>
> **[About the typo]**
> Thanks for your advice! We fix the typo in the new version of the paper.
>
> > *References:*
> > *[1] Mao, Xudong, et al. "Least squares generative adversarial networks. "*

---

### Official Review · AnonReviewer2 · 2020-10-28
**Non-AR end-to-end TTS model with controllability**

**Rating:** 7
**Confidence:** 5

**Review:**

This paper further extends the non-autoregressive end-to-end TTS model "FastSpeech".  The new work includes (1) removing the knowledge distillation from an AR model to the non-AR model which was required in FastSpeech, (2) introduced "variance" representations such as energy and F0 in addition to duration.  (1) made the training procedure simpler.  (2) improved the naturalness as well as controllability of synthesized speech.

Technical details and model architecture are available in the body of the paper, whereas additional details are also provided in the appendix.  The model was evaluated in the LJSpeech dataset, which is a publicly available dataset and often used for TTS model evaluations.  Audio samples are also provided in the demo page.

There are a few parts need to be addressed.  (1) made the training procedure simpler but it also introduces the necessity to rely on external aligner, which actually complicates the whole voice building process.  It is not well discussed in the paper.  (2) also introduces the dependency to external modules, such as F0 extractor and makes the model less "end-to-end".  The dependency to an external F0 extractor can introduce errors, as F0 extractors sometimes produce errors such as pitch halving/doubling.  The authors should also discuss it from this point.

Comments:
- For pitch and energy 256-bin quantization was used.  Did the author observe any issue if continuous values were directly used?
- It is unclear whether there is any difference between FastSpeech 2s waveform decoder and Parallel WaveGAN decoder.
- Comparison with FastPitch: the authors claimed "results in better prosody" but I cannot see any evidence or experimental validation for this claim.  If you have any results, please include it.  Otherwise, this is too strong claim.
- To demonstrate FastSpeech 2 comparable to AR models, having a side-by-side comparison between them is helpful.
- Table 3 includes moments of pitch of natural and synthetic speech.  Can you include these numbers of FastSpeech 2 with F0 fit via MSE which you discussed in the ablation study?
- If I understand correctly pitch mean/var are constant within an utterance, but pitch spectrogram is frame-by-frame.  In Figure 2 both of them are predicted from pitch predictor but the nature of these values are different (global / frame-by-frame).  Please include more explanation how they are predicted in the pitch predictor.

---

> ### Author Response · Authors · 2020-11-22
> **Response to AnonReviewer2**
>
> Thanks for your comments on our paper. We reply to your questions as follows:
>
> **[About the training procedure and the dependency to external modules]**
> To ensure high quality of FastSpeech 2, we use an external high-performance alignment tool and pitch extraction tools, which may seem a little complicated, but are very helpful for high-quality and fast speech synthesis. High quality, fast and fully end-to-end training without any external libraries is definitely the ultimate goal of neural TTS and also a very challenging problem. To avoid complicating the TTS model architecture, we indeed introduce some external tools. We believe there will be a simpler pipeline to achieve this goal in the future and we will certainly work on fully end-to-end TTS without external models. We add the above discussions about the training procedure in Section 4.
>
> **[About errors introduced by the external F0 extractor]**
> Since we cannot obtain the ground-truth pitch contour of LJSpeech dataset, we choose some samples and plot the pitch contour on the mel-spectrogram. We can judge that most of the frame-level pitch is correct according to the formants of the mel-spectrogram. And the F0 frame error (FFE) of our pitch extraction algorithm (DIO) is 7.1% reported in [1] which is acceptable.
>
> **[About the f0 and energy quantization]**
> We conduct the CMOS evaluation and the results show that the voice quality drops ($-0.26$ CMOS) if continuous values are directly used.
>
> **[About the waveform decoder]**
> The structure of the waveform decoder in FastSpeech 2s is nearly the same as Parallel WaveGAN decoder. A minor difference is that the condition input of the former is hidden states while that of Parallel WaveGAN decoder is mel-spectrograms.
>
> **[About the comparison with FastPitch]**
> By claiming “results in better prosody”, we mean that FastSpeech 2 and 2s can better predict the variations in pitch better when applying continuous wavelet transform, rather than comparing with the concurrent work, FastPitch. We modify this claim in the new version of the paper to avoid misunderstanding.
>
> **[About the side-by-side comparison]**
> We add the side-by-side comparison (CMOS) between FastSpeech 2 and Transformer TTS to Table 1 in the new version of the paper. The CMOS results show that FastSpeech 2 outperforms Transformer TTS and FastSpeech.
>
> **[About the moments of FastSpeech 2 with F0 fit via MSE]**
> We add the moments of FastSpeech 2 with F0 fit via MSE (denoted as FastSpeech w/o CWT) in Table 3 of the new version of the paper and analyze the results in Section 3.2.3. The results show that CWT is helpful for pitch modeling.
>
> **[About the prediction of mean/variance of pitch contour]**
> To predict the mean/variance of pitch contour for each utterance, we first average the hidden states output by the 1D-convolutional network in the pitch predictor on the time dimension to a global vector, and then project it to mean/variance using a linear layer. To predict the pitch spectrogram, we directly feed the frame-level hidden states to another linear layer. We add these details to Appendix C.2 in the new version of the paper.
>
> > *References:*
> > *[1] Drugman, Thomas, et al. "Traditional machine learning for pitch detection."*

---

### Official Review · AnonReviewer4 · 2020-10-29
**Clear improvement over FastSpeech**

**Rating:** 8
**Confidence:** 4

**Review:**

This paper describes a number of improvements over the non-autoregressive FastSpeech TTS model.

One of the trends that TTS has seen since Tacotron was announced is a retreat from fully end-to-end modeling for TTS.  FastSpeech2 is another entry in that.  The text input is first converted to phones, and prosodic information can be inferred from text (in a unified system) or can be conditioned by a user.  All of these are valuable improvements for quality and customizability.

In particular, FastSpeech takes an interesting modular approach to conditioning on pitch, duration and energy for its prosody component.  It relies on ground truth predictors during training, but learns to infer these via multitask training so they are not required during inference.   The use of a dense and continuous pitch representation as a pitch spectrogram is interesting as well.  This could be a limitation for customizability as a user would be more comfortable to provide a pitch target rather than specifying a pitch spectrogram.

This paper is clearly written and includes sufficient detail for reproducibility.  The evaluation is convincing especially as it relates to training and inference time.

Minor comment: In Table 2 the FastSpeech2s training time number is omitted.  This is because it is not directly comparable to FastSpeech2 or the other training numbers -- it includes training a neural vocoder while the others do not.  However, since the discussion of training time as a valuable measure is already begun, I would suggest that the authors include this number, with the explanation that it is not comparable. (or include vocoder training time along with the other methods.)

Comment: The one-to-many TTS problem --- where one lexical utterance can be produced in many different correct ways -- is only partly addressed here for two reasons. 1) there is variation other than duration, energy and pitch that can vary between realizations of the same utterance, voice quality, background and channel noise, and speaker effects.  2) more significantly, the one-to-many problem is only approached during model training.  During inference, the process is still deterministic given the state of the model.    The inference side of the problem is important for objective model evaluation.

---

> ### Author Response · Authors · 2020-11-22
> **Response to AnonReviewer4**
>
> Thanks for your comments on our paper. We reply to your questions as follows:
>
> **[About the training time of FastSpeech 2s]**
> Thanks for your advice! We add the training time of FastSpeech 2s in Table 2 in the new version of the paper.
>
> **[About the one-to-many TTS problem]**
> Yes, we will certainly add more variance information to variance adaptor in the future and we take pitch and energy as examples.  In this work, we mainly focus on the training procedure of non-autoregressive TTS: to simplify the output distribution assumption and ease the one-to-many mapping issue. In the future, we can certainly generate the parameters of the output distribution and sample the output mel-spectrogram from it to make the results non-deterministic.

---

### Official Review · AnonReviewer5 · 2020-11-06
**New improvements to the FastSpeech model that lead to improved quality and  performance**

**Rating:** 7
**Confidence:** 5

**Review:**

This paper is motivated by, and intends to address shortcomings of, the previously published FastSpeech paper, specifically problems with the internal duration prediction of that model, and its requirement for a teacher-student configuration. The submission makes clear what the problems with that original approach are, and the evaluation later confirms this. What is missing from this work, however, is a more candid statement about the trade-offs between the FastSpeech and FastSpeech2 architectures since the current implementation requires externally aligned datasets, or the resources to develop them.


In other words, although the authors present this paper as a natural and organic extension of the original FastSpeech work, one could argue that in some ways it represents a step backwards with respect to the spirit of sequence-to-sequence models that solve the alignment problem without the need for external supervision. One may very well have the resources to afford such supervision, so there's definitely a place for this type of model, but the paper should be more forthcoming in the pros and cons of this new approach.

Having a duration-prediction model explicitly trained with externally-determined duration targets, in fact, brings this work closer to the more "classical" earlier-generation pipeline neural architectures (pre seq-to-seq), with the notable difference that everything here is trained jointly. I think the paper would be well served by a broader review of the literature that tries to situate it beyond its similarity with FastSpeech only. Some relevant parts of the literature seem uncited (I've included examples below). If space is an issue, some of the material in the Appendix could be shortened, or just referenced.


The paper is generally well written and easy to follow. Some comments and needed clarifications follow:

The paper says that the variance adaptor "adds" different variance information. Do you literally mean that you sum up the different embeddings (the use of a + in Fig. 1b would suggest as much), or do you concatenate/stack them? If it's the former, try to motivate this choice since summing up linguistic embeddings with acoustic embeddings doesn't necessarily seem an obvious choice.

Sec. 2.3 --  *Previous neural network based TTS systems with pitch prediction (Arik et al.,2017; Gibiansky et al., 2017) often predict pitch contour directly.* -- There is, in fact, an extensive (and relevant, but uncited) literature on direct neural prediction of f0 with external f0 targets as the authors do here. See, e.g,:
- Statistical Parametric Speech Synthesis Using Deep Neural Networks (Zen et al., 2013)
- Prosody Contour Prediction with Long Short-Term Memory, Bi-Directional, Deep Recurrent Neural Networks (Fernandez et al., 2014)
- TTS Synthesis with Bidirectional LSTM based Recurrent Neural Networks (Fan et al., 2014)

The idea of conditioning seq-2-seq models on some acoustic properties of the output (in the spirit of the variance adaptor of FastSpeech2) is also not new, and has been proposed to either improve control or stability of such models. See. e.g.,

- Sequence to Sequence Neural Speech Synthesis with Prosody Modification Capabilities (Shechtman et al., 2019)
and more recently:
-  Controllable Neural Text-to-Speech Synthesis Using Intuitive Prosodic Features (Raitio et al., 2020)


 Sec. 2.5: *Previous non-autoregressive vocoders (Oord et al., 2017; Prenger et al., 2019; Yamamoto et al.,2020; Kumar et al., 2019) are not complete text-to-speech systems, since they convert time aligned linguistic features to waveforms, and require a separate linguistic model to convert input text to linguistic features or an acoustic model to convert input text to acoustic features (e.g., mel-spectrograms). FastSpeech 2s is the first attempt to directly generate waveform from phoneme sequence, instead of linguistic features or mel-spectrograms.* -- A phoneme sequence is a type of linguistic information that needs to be extracted from the input text. It's hard to argue that something that requires a G2P is more of a full TTS system. I assumed that the input text was also normalized prior to the G2P, so by starting from a phoneme sequence, you're assuming access to both a normalizer and a G2P.


Evaluation; What type of sentences were used in the evaluation? The paper mentions that the training of FastSpeech2s required using shorter clips to deal with memory issues. Do you restrict your listening test to also contain only simpler, shorter sentences? Systems that perform close to each other when synthesizing shorter prompts can behave very differently when attempting more complex inputs (questions; lists; sentences with syntactic embedding or relative clauses; etc.)

Though the numbers cluster nicely, consider adding parwise significance tests to the results in Table 1.

The analysis/resynthesis condition (GT(Mel+PWG)) shows a drop of 0.38 MOS points with respect to the ground truth (4.3 vs 3.92). Can you comment on this? It seems large.

Sec. 2.2.3. Ablation studies -- Any reason why you didn't include similar tables to Table 6 for the other studies (2nd and 3rd paragraph in this section)?
Minor edits:

- since waveform contains --> since the waveform contains
- discuss the challenges lie --> discuss the challenges that lie
- pitch predictor consists --> the pitch predictor consists
- A few acronyms incorrectly capitalized in the bibliography (tts --> TTS / flowavenet --> FloWaveNet / etc.)
- It's diffcult to make sense out of the Ryan, 1994 reference.

---

> ### Author Response · Authors · 2020-11-22
> **Response to AnonReviewer5**
>
> Thanks for your comments on our paper. We reply to your questions as follows:
>
> **[Discussions about cons of our method]**
> Thanks for your advice! We add some discussions about this in Section 4 in the new version of the paper. To ensure high quality of FastSpeech 2, we use an external high-performance alignment tool and pitch extraction tools, which may seem a little complicated, but are very helpful for high-quality and fast speech synthesis. High quality, fast and fully end-to-end training without any external libraries is definitely the ultimate goal of neural TTS and also a very challenging problem. To avoid complicating the TTS model architecture, we indeed introduce some external resources and tools. We believe there will be a simpler pipeline to achieve this goal in the future and we will certainly work on fully end-to-end TTS without external models.
>
> **[About “adding different variance information”]**
> We just sum these embeddings together rather than concatenate them for simplicity. We conduct CMOS evaluation to compare models using “add” and “concatenate” when fusing variance information and the results show that there is no significant difference between them.
>
> **[About the missing references on pitch prediction]**
> Compared with these works with pitch prediction, FastSpeech 2 and 2s adopt joint-trained self-attention based feed-forward network to generate mel-spectrograms or waveform in parallel. We also introduce CWT to improve the pitch prediction accuracy. Compared with those works focusing on controllable speech synthesis, our work aims to solve the one-to-many mapping issue by adding some variance information input. As a byproduct, we also achieve controllable generation. We cite these works in the new version of the paper.
>
> **[About the end-to-end claims]**
> Yes, we need another text normalizer and G2P model to convert text to phoneme. Compared with phoneme we used, the linguistic features widely used in vocoders contain more information such as context and duration. In this paper, our end-to-end claims refer to phoneme-to-waveform and we add a footnote to make it clear.
>
> **[About the type of sentences used in the evaluation]**
> We used the whole sentence in LJSpeech for evaluation without clipping. We randomly clip audio in training for FastSpeech 2s. But in Inference, FastSpeech 2s generates the whole long sentence as same as other TTS models (e.g., FastSpeech 2 and Transformer TTS).
>
> **[About the pairwise significance tests]**
> Thanks for your advice, we add more pairwise results (CMOS) to Table 1 in the new version of the paper. The CMOS results also show that FastSpeech 2 outperforms Transformer TTS and FastSpeech in voice quality.
>
> **[About the GT MOS gap]**
> The MOS gap of between ground-truth and resynthesis audio matches that in the original paper of Parallel WaveGAN *[1]*: our gap is $4.30-3.92=0.38$ and that in Parallel WaveGAN paper is $4.46-4.06=0.40$.
>
> **[About omitted tables in ablation studies]**
> Because there is only one number (CMOS) for each paragraph, and due to the limitation of the space, we do not include the table and put the number in the text directly.
>
> **[About typos]**
> We’ve fixed these typos in the new version of the paper. The case format of these acronyms in the bibliography is specified by the conference template. Ryan (1994) et al. analyze the characteristic of Ricker wavelet, also known as mexican hat wavelet, which we used for CWT.
>
> > *References:*
> > *[1] Yamamoto, Ryuichi, Eunwoo Song, and Jae-Min Kim. "Parallel WaveGAN: A fast waveform generation model based on generative adversarial networks with multi-resolution spectrogram."*

---

### Official Review · AnonReviewer1 · 2020-11-10

**Rating:** 5
**Confidence:** 5

**Review:**

This work presents several improvements over the original teacher-student framework in FastSpeech: 1) training the model with ground-truth target instead of the output from teacher, 2) extract phoneme duration, pitch and energy from speech and directly take them as conditional inputs in training and use predicted values in inference.  Importantly, it uses pretrained forced aligner (MFA) to extract the phoneme durations for training.

Comments:

1, Typo: In introduction, "alleviate the one-to mapping"

2, The "one-to-many mapping" is not an issue in general. This setting widely exists in generative modeling, e.g., label conditioned image synthesis, mel spectrogram conditioned waveform synthesis. The problem of FastSpeech (and other non-autoregressive TTS models) is really due to the over-simplified output distributions, which assume conditional independence between frames and frequency bins for mel spectrogram. As a result, these models doesn't account for the variations in real data.

3, The introduction, section 2.1, and section 2.2 contain too much duplicated information. One may shorten the text properly.

4, The motivation & architecture of FastSpeech 2s (Figure 1 (a)) is similar to the previous text-to-waveform model clarinet. Both of them use mel prediction task to guide the training. The authors overclaim that "FastSpeech 2s is the first attempt to directly generate waveform from phoneme sequence".

5, The model is similar to traditional TTS pipeline with separate duration model, pitch/F0 model etc. The CWT/iCWT-based pitch predictor is interesting.

6, The MOS of autoregressive Tacotron 2 is relatively low. Which implementation did you use?

7, One may also report the standard deviation, skewness, and kurtosis of pitch in synthesized audio from autoregressive model (Tacotron 2 and Transformer TTS). I assume their results would be closer to GT.

8, In Table 5, the durations from teacher model are extracted by teacher forcing ground-truth mel spectrogram? One may mention that MFA is pretrained on a much larger dataset, thus may have better generalization than the teacher model trained on small LJSpeech dataset.

Pros:
- Good sample quality.
- Sufficient ablation study.

Cons:
- The proposed pipeline is far more complicated than existing end-to-end TTS model.
- Inaccurate & confusing claims (see my comments).
- The novelty is rather limited.

---

> ### Author Response · Authors · 2020-11-22
> **Response to AnonReviewer1**
>
> Thanks for your comments on our paper. We reply to your questions as follows:
>
> **[About the typo]**
> We fix this typo in the new version of the paper.
>
> **[About the one-to-many mapping issue]**
> Yes, you are right. One-to-many mapping exists in most generative modeling tasks. However, the one-to-many mapping issue in non-autoregressive generation is more severe than that in autoregressive generation. For example, in training autoregressive TTS, the current speech frame can condition on the previous ground-truth speech frames, which can reduce the uncertainty in prediction. In this work, since we cannot use previous ground-truth speech frames for prediction in non-autoregressive TTS, we instead use more input information (e.g., pitch, energy and more accurate duration) to help predict the variations in speech, which can reduce the uncertainty of the conditional distribution of output mel-spectrograms given input. In this way, we can use simple loss (L1 or L2) to model the output distribution rather than other complicated losses or model architecture (e.g., GAN and normalizing flow). In the future, we will try to model the joint distribution of mel-spectrograms.
>
> **[About the duplicated information in the three sections]**
> We remove the redundant part in both sections in the new version of the paper.
>
> **[About the “first attempt” claim]**
> Since Clarinet has an autoregressive decoder and does not generate waveform fully in parallel, we modify the claim to “FastSpeech 2s is the first attempt to directly generate waveform from phoneme sequence fully in parallel” in the new version of the paper.
>
> **[About the Tacotron 2 implementation]**
> We use the open-source implementation of [ESPNet](https://github.com/espnet/espnet) *(model configuration: `train_pytorch_tacotron2.v3.yaml`)*. The MOS gap between Tacotron 2 and Transformer TTS in our paper ($3.70-3.72=-0.02$) matches that in *[1]* ($4.20-4.25=-0.05$).  And the MOS gap between Tacotron 2 and GT in our paper ($3.70-4.30=-0.60$) also matches that in the concurrent work, Glow-TTS *[2]* ($3.88-4.54=-0.66$), which is larger than those in some works *[3,4]* using WaveNet vocoder *[5]*, since WaveNet vocoder achieves higher naturalness of generated speech than Parallel WaveGAN we used.
>
> **[About pitch moments of autoregressive models]**
> We’ve updated the pitch moments of Tacotron 2 and Transformer TTS in the new version of the paper. We can see that pitch moments of FastSpeech 2 and 2s are closer to GT than Tacotron 2 and Transformer TTS, which is consistent with MOS results in Table 1 (FastSpeech 2 and 2s achieve higher MOS than Tacotron 2 and Transformer TTS).
>
> **[About the durations in Table 5]**
> The durations from the teacher model are extracted from the attention map in free-running mode rather than teacher-forcing.
>
> **[About the training data of MFA]**
> Our MFA is trained on the LJSpeech dataset only and therefore it doesn't make an unfair comparison.  We clarify them in the new version of the paper.
>
> **[About the pipeline]**
> In this work, we mainly focus on addressing the issues of FastSpeech and solving the one-to-many mapping problem in non-autoregressive TTS. To ensure high quality of FastSpeech 2, we use an external high-performance alignment tool and pitch extraction tools, which may seem a little complicated, but are very helpful for high-quality and fast speech synthesis. High quality, fast and fully end-to-end training without any external libraries is definitely the ultimate goal of neural TTS. In the future, we will certainly work on fully end-to-end TTS without external models. We add some discussions about this in Section 4 in the new version of the paper.
>
> **[About the novelty]**
> The main contributions of this paper include: 1) we analyze one-to-many mapping and information gap in non-autoregressive TTS, which are important for us to solve the problem in non-autoregressive TTS. To the best of our knowledge, few works have analyzed this kind of problem in non-autoregressive TTS before. 2) We introduce variation information of speech (e.g., pitch, energy and more accurate duration) into FastSpeech to reduce the information gap between text and speech and alleviate the one-to-many problem, which is verified to be very effective. 3) We introduce CWT to pitch modeling to further improve the pitch accuracy. 4) Benefiting from the reduction of information gap, as a byproduct, we propose a directly and parallel text-to-waveform model to further simplify the training and inference.

---

> > ### Author Response · Authors · 2020-11-22
> > **References**
> >
> > > *References:*
> > > *[1] Hayashi, Tomoki, et al. "Espnet-TTS: Unified, reproducible, and integratable open source end-to-end text-to-speech toolkit."*
> > > *[2] Kim, Jaehyeon, et al. "Glow-TTS: A Generative Flow for Text-to-Speech via Monotonic Alignment Search."*
> > > *[3] Shen, Jonathan, et al. "Natural TTS synthesis by conditioning wavenet on mel spectrogram predictions."*
> > > *[4] Li, Naihan, et al. "Neural speech synthesis with Transformer network."*
> > > *[5] Oord, Aaron van den, et al. "WaveNet: A generative model for raw audio."*

---

### Decision · Program_Chairs · 2021-01-07
**Final Decision**

**Decision:**

Accept (Poster)

**Comment:**

This paper presents a number of techniques to improve the existing non-autoregressive end-to-end TTS model -- FastSpeech. These techniques include replacing the teacher forcing with ground truth targets and using a variation adaptor to introduce auxiliary information such as duration, energy and pitch.  The experiments show that the proposed Fastspeech 2 model is faster in training  compared to the existing FastSpeech model and meanwhile can still achieve high quality synthesized speech.  The work reported in the paper is essentially about system improvement over FastSpeech but has it value in the speech community given the current interest in non-autoregressive rapid TTS.  On the other hand, concerns are also raised regarding the complexity of the pipeline and the significance of the novelty. The authors' rebuttal is good and has addressed most of the concerns.  Overall, this is an interesting paper and can be accepted.